# Recent Advances in Understanding Nrf2 Agonism and Its Potential Clinical Application to Metabolic and Inflammatory Diseases

**DOI:** 10.3390/ijms23052846

**Published:** 2022-03-05

**Authors:** Min-Ji Kim, Jae-Han Jeon

**Affiliations:** 1Department of Endocrinology in Internal Medicine, Kyungpook National University Hospital, Daegu 41944, Korea; irisfullip@naver.com; 2Department of Internal Medicine, School of Medicine, Kyungpook National University, Kyungpook National University Chilgok Hospital, Daegu 41404, Korea

**Keywords:** Nrf2, oxidative stress, metabolic disease, inflammatory disease, clinical trials

## Abstract

Oxidative stress is a major component of cell damage and cell fat, and as such, it occupies a central position in the pathogenesis of metabolic disease. Nuclear factor-erythroid-derived 2-related factor 2 (Nrf2), a key transcription factor that coordinates expression of genes encoding antioxidant and detoxifying enzymes, is regulated primarily by Kelch-like ECH-associated protein 1 (Keap1). However, involvement of the Keap1–Nrf2 pathway in tissue and organism homeostasis goes far beyond protection from cellular stress. In this review, we focus on evidence for Nrf2 pathway dysfunction during development of several metabolic/inflammatory disorders, including diabetes and diabetic complications, obesity, inflammatory bowel disease, and autoimmune diseases. We also review the beneficial role of current molecular Nrf2 agonists and summarize their use in ongoing clinical trials. We conclude that Nrf2 is a promising target for regulation of numerous diseases associated with oxidative stress and inflammation. However, more studies are needed to explore the role of Nrf2 in the pathogenesis of metabolic/inflammatory diseases and to review safety implications before therapeutic use in clinical practice.

## 1. Introduction

Nuclear factor-erythroid-derived 2-related factor 2 (Nrf2) is a key transcription factor that regulates cellular homeostasis in response to oxidative stress [1]. Within the cytosol, Nrf2 is suppressed by Kelch-like ECH-associated protein 1 (Keap1), a substrate adaptor protein [2]. Under conditions of cellular stress, the cysteine thiols of Keap1 are modified, thereby preventing degradation of Nrf2 and allowing its translocation to the nucleus, where it binds to its target genes, which include those encoding antioxidant and detoxification enzymes [3,4]. Accordingly, pharmacological agents that target Nrf2 are being developed to protect tissues from oxidative damage and prevent progression of a number of diseases [5,6,7,8]. In this review, we focus on the emerging role of Nrf2 agonism during regulation of multiple metabolic/inflammatory diseases, including diabetes, obesity, inflammatory bowel disease, and autoimmune diseases. We also summarize recently developed Nrf2-targeting drugs and relevant ongoing clinical trials.

## 2. Regulation of Nrf2

### 2.1. The Seven Domains of Nrf2 and Their Functions

Nrf2 is a basic leucine zipper (bZIP) transcription factor with a 66 kDa cap’n’collar (CNC) structure [9,10]. Expression of Nrf2 is observed in many tissues that are exposed to the external environment (i.e., the gastrointestinal tract and upper respiratory system), as well as metabolically active organs (i.e., endocrine tissues, heart, skeletal muscle, and brain) and detoxification organs (i.e., the liver and kidneys) [11]. Nrf2 contains seven domains, named Nrf2-erythroid cell-derived protein with CNC homology (Neh) domains (Figure 1) [12]. The Neh1 domain is a CNC-bZIP domain that allows Nrf2 to heterodimerize with small musculoaponeurotic fibrosarcoma (sMaf) proteins (Maf F, G, and K) [13,14]. The Neh2 domain allows Nrf2 to bind to its cytosolic repressor, Keap1 [15]: seven lysine residues within the Neh2 domain (upstream of the ETGE motif) are degrons that are essential for Keap1-dependent polyubiquitination and degradation of Nrf2 [16]. The Neh3 domain stabilizes the Nrf2 protein and acts as a transactivation domain during transcription [17]. The Neh4 and Neh5 domains also act as transactivation domains, but bind to cyclic adenosine monophosphate (cAMP) response element binding protein, which possesses intrinsic histone acetyltransferase activity [15]. The Neh6 domain contains another degron involved in the redox-insensitive process of Nrf2 degradation [18]; it has two motifs that interact with β-transducin repeat-containing protein and is responsible for β-transducin repeat-containing protein-mediated proteasomal degradation [19]. This occurs even in stressed cells, in which the half-life of the Nrf2 protein is normally extended relative to unstressed conditions by suppression of other degradation pathways [20]. Finally, the Neh7 domain mediates repression of Nrf2 by retinoid X and retinoic acid receptors, which prevent binding of transcription co-activators to the Neh4 and Neh5 domains [21].

### 2.2. The Keap1–Nrf2 System in Regulating Nrf2

Nrf2 activity is regulated by a number of mechanisms, including the ubiquitin proteasome degradation system (UPS) [22], post-translational modifications [23,24,25,26,27], epigenetic regulation [28,29], microRNAs [30,31], and autoregulation [32,33]. The main regulatory pathway in response to agonists is the Keap1–Nrf2 UPS system (Figure 1). Keap1 is a member of the BTB-kelch protein family, all of which contain two distinct domains within their structure: BTB (Broad Complex, Tramtrack and Bric-à-Brac) and a Kelch domain. BTB (aa 50–179) is a homo-dimerization Cullin 3 (Cul3) binding domain that contains the Cys^151^ residue. IVR (an intervening linker region at aa position 180–314) contains two critical cysteine residues, Cys^273^ and Cys^288^. The DGR (double-glycine repeat)/Kelch domain (aa position 327–611) contains an Nrf2 binding domain (see the left side panel in Figure 1). The BTB, IVR, and Kelch domains within Keap1 are important for regulating Nrf2. The Keap1 homodimer binds to the ETGE and DLG motifs of the Neh2 domain of Nrf2 [34,35]. The high-affinity ETGE motif acts as the “Hinge”, and the DLG motif (which has 200-fold weaker affinity) acts as the “Latch” [36,37,38]. The DLG motif, which is located in the N-terminal region, is important for ubiquitination and degradation of Nrf2 [18,22], while the ETGE motif is essential for interaction with Keap1 [39]. Keap1 acts as a sensor for Nrf2; under basal conditions, when Keap1 binds to Nrf2, Nrf2 is ubiquitinated by the Keap1–Cul3 complex and is then degraded by the 26S proteasome [40]. In 2021, Yamamoto’s group, which originally proposed the Hinge-Latch model, conducted an elegant nuclear magnetic resonance spectroscopy study and developed an updated Hinge-Latch model [38]. This study confirmed the Hinge-Latch mechanism by monitoring accumulation of the autophagy chaperone P62 and non-electrophilic inducers (Keap1-Nrf2 protein-protein inhibitors PRL295 and NG262). Most Nrf2 agonists, which are electrophilic inducers, do not appear to block Nrf2 ubiquitination through the Hinge-Latch mechanism; rather, they block it by causing conformational changes in Keap1 through cystine modification [38]. When cells are exposed to oxidative stress, Nrf2 escapes the negative regulation of Keap1, and the Keap1 sensor permits Nrf2 entry to the nucleus. After Nrf2 translocates to nucleus, it heterodimerizes with sMaf proteins [41], allowing binding to the cis-regulatory element or enhancer sequence [42]. Target gene expression is induced by binding of the resulting heterodimer to cis-acting consensus DNA sequences referred to as antioxidant responsive elements (AREs) or electrophile response elements. These sequences are now collectively called CNC–sMaf binding elements [43]. The Nrf2–sMAF complex activates transcription of cytoprotective genes such as heme oxygenase-1, superoxide dismutase, NAD(P)H:quinone oxidoreductase-1, and glutathione cysteine ligase [44]. The impact of the Nrf2–ARE pathway has been investigated using Nrf2 and Keap1 knockout mouse models, which exhibit altered responses to oxidative stress and toxicity [45]. Through the Nrf2–Keap1 system, Nrf2 regulates expression of cytoprotective genes involved in antioxidant and detoxification systems, catalytic activity, glutathione processing, glucose metabolism, lipid metabolism, and mitochondrial function [46,47,48,49,50,51,52,53].

The mechanisms that activate Nrf2 mentioned so far depend on direct interaction with Keap1. However, a number of recent studies identified Keap1-independent Nrf2 regulatory mechanisms, including phosphorylation by protein kinase C (PKC) [23], phosphoinositide 3-kinase (PI3K)/protein kinase B (Akt) [54], and glycogen synthase kinase 3 β (GSK-3β) [55]. In addition, the presence of an ARE-like sequence in the promoter region of the Nrf2 gene provides evidence of Nrf2 autoregulation [56]. Nrf2 can self-activate its own gene expression to increase the production of Nrf2 protein. Several studies have shown that miRNA, another epigenetic mechanism, also regulates Nrf2. For example, direct targeting of miRNA (e.g., miR-200a, miR-7, miR-141, miR-432) of Keap1 is known to keep Nrf2 free from Keap1, leading to Nrf2 activation [57]. Nrf2 is repressed by β-transducin repeat-containing protein (β-TrCP) though its GSK-3β-catalysed phosphorylation of the Neh6 domain, which is responsible for ubiquitination of Nrf2, another Keap1 independent pathway [58]. Also, Nrf2 is activated by several oxidative stress-independent pathways. For example, P62, a ubiquitin-binding protein, competes with Nrf2 for binding to Keap1. Accumulation of P62 disrupts the DLG-Keap1 interaction and releases Nrf2. In addition, phosphorylated Nrf2 dissociates from Keap1 and translocates to the nucleus [56].

Thus, Nrf2 influences several important metabolic processes apart from antioxidant metabolism; these include glucose [46] and lipid metabolism [59], iron metabolism [49,60], and anti-inflammatory responses.

## 3. Role of Nrf2 Agonism in Chronic Inflammatory Diseases

### 3.1. Role Nrf2 in Immune Cells and in Production of Inflammatory Mediators

#### 3.1.1. The Role of Nrf2 in Immune Cell Function

Inflammation results in production of ROS, which can lead to significant activation of Nrf2 [19]. The immunomodulatory role of Nrf2 affects the Nrf2–Keap1 pathway and may protect host cells from a variety of inflammatory disorders such as obesity, metabolic syndrome, and autoimmune disease. However, there are many questions surrounding whether the Nrf2–Keap1 pathway is always unidirectional; the answer to this question is that it appears to be immune cell type-dependent [11]. For example, studies in a septic mouse model show that increased Nrf2 expression in *Keap1*^−/−^ M1 macrophages suppresses expression of proinflammatory genes and subsequent tissue injury [61]. From another perspective, disruption of Keap1 in murine myeloid leukocytes increases the bacterial phagocytic activity of peritoneal macrophages [62]. In addition, *Nrf2*-deficient (*Nrf2*^−/−^) peritoneal neutrophils show increased expression of TNF-α, IL-6, monocyte chemoattractant protein-1, and macrophage inflammatory protein-2 [63]. Activation Nrf2 in T cells by the Nrf2 activator tert-butylhydroquinone impairs Th1-induced inflammatory responses and biases T cells towards Th2 differentiation and an anti-inflammatory response [64]. Nrf2 activation in the CD4-Keap1-KO mouse model of acute kidney injury promotes and enhances activation and expansion of Treg cells [65]. By contrast, Nrf2 deficiency increases oxidative damage, which induces differentiation of Th17 cells, which are themselves associated with a proinflammatory response [66]. Finally, Nrf2 activation in myeloid-derived suppressor cells (MDSCs) leads to their expansion, which inhibits T cell responses [67]. These pathways are summarized in Figure 2 and Figure 3.

#### 3.1.2. The Inhibitory Role of Nrf2 in Production of Inflammatory Mediators (Cytokines and Proteases)

Cytokines are hallmark mediators of inflammation; representative examples are interleukins (ILs), interferons (IFNs), tumor necrosis factor (TNF), and chemokines. Accumulating evidence suggests that Nrf2 protects against the effects of inflammatory cytokines (see recent review articles [73,74,75]). 

Increased expression of nuclear factor κ-light-chain enhancer of activated B cells (NF-κB) is central to inflammatory responses, and its activity is dependent on the redox status of the cell [74]. ROS and reactive nitrogen species stimulate and exacerbate inflammatory responses that are mechanistically linked with the p65 subunit of NF-κB [76]. In quiescent immune cells, NF-kB interacts physically with the nuclear κ-B inhibitor (IκBα), which prevents nuclear localization of the former; however, a proinflammatory milieu activates IkB kinase (IKK)β, which phosphorylates IκBα. Phosphorylated IκBα is targeted for degradation, thereby removing it from NF-κB and allowing its nuclear translocation [77]. In terms of the relationship between NF-κB and Nrf2, IKKβ (like Nrf2) possesses an ETGE motif that can bind Keap1 to target it for ubiquitination and proteasomal degradation. Under conditions of ROS abundancy, Keap1 is inhibited, thereby stabilizing IKKβ. This in turn leads to phosphrylation and degradation of IκBα, which results in aberrant induction of NF-κB [78].

The interplay between Nrf2 and NF-κB is complex [74]. For example, NF-κB transcriptionally activates expression of Nrf2 at a specific κB promoter site [79]. In addition, NF-κB and Nrf2 compete for binding to the coactivator histone acetyltransferase CREB-binding protein (CBP)/p300. In other words, an abundance of NF-κB disrupts the interaction between Nrf2 and the CH1-KIX domain of CBP [80]. The same study also revealed that NF-κB promotes recruitment of the corepressor histone deacetylase 3 (HDAC3) to induce Nrf2-driven expression of ARE [80]. NF-κB also binds to Keap1 and translocates it to the nucleus, thereby favoring ubiquitination and degradation of Nrf2. NF-κB signaling inhibits the Nrf2-ARE pathway through the interaction between p65 and Keap1 [81]. 

Likewise, Nrf2 inhibits NF-κB activity via a number of cellular mechanisms. NRF2 inhibits NF-κB activity indirectly by suppressing ROS. Furthermore, lipopolysaccharide (LPS) activates a fast, proinflammatory NF-κB response but a slow NRF2 response. Chronologically, NF-κB activity is inhibited when NRF2 is most active [82]. Importantly, NRF2-deficient cells showed increased expression of p65-NF-κB protein, although mRNA levels remain unchanged, indicating post-translational modifications. Consistent with this, Nrf2-deficient mouse embryonic fibroblasts show greater activation of NF-κB in response to LPS [83]. A very recent study corroborates these findings. That study shows that proinflammatory cytokines induced by LPS through the TLR4/NF-κB pathway promote expression of Nrf2, followed by its translocation to the nucleus. Strikingly, it also shows that Nrf2 inhibits expression of proinflammatory cytokines by binding to p65 directly [84]. Traditionally, Nrf2 activators reduce ROS levels and inhibit ROS-mediated activation of NF-κB and NF-κB-depedent inflammatory mediators (i.e., IL-1β, IL-6, TNF-α, and COX-2) [85], whereas others reduce expression of inflammatory mediators without affecting NF-κB expression or activity [86]. 

The aforementioned M1-polarized macophages promote secretion of proinflammatory cytokines such as TNF, IL-1, and IL-6 [87]. Nrf2 suppresses inflammatory responses by macrophages by blocking transcription of proinflammatory cytokines [61]. In line with this, administration of itaconate, a Nrf2-activating metabolite, inhibits the STING/NF-κB axis in chondrocytes and promotes M2 polarization of macrophages, thereby alleviating osteoarthritis (OA) [88]. The mechanism by which itaconate activates Nrf2 was identified by O’Neill [89]. Essentially, itaconate alkylates cysteine residues 151, 257, 288, 273, and 297 of Keap1, thereby enabling Nrf2 to increase expression of antioxidant and anti-inflammatory mediators [89]. The protective role of itaconate derivatives was further confirmed in the context of disease. A study shows that 4-octyl itaconate activates Nrf2 in patients with systemic lupus erythematosus, thereby inhibiting proinflammatory cytokine production by peripheral blood mononuclear cells [90]. Dimethyl itaconate suppresses LPS-induced production of TNF-α, IL-6, and NOS2 by bone marrow-derived macrophages (BMDMs). Administration of dimethyl itaconate increases survival rates, decreases serum levels of TNF-α and IL-6, and ameliorates lung injury in septic mice. Importantly, survival rates of dimethyl itaconate-treated septic Nrf2^−/−^ mice were much lower than wild-type littermates, suggesting that the role of dimethyl itaconate is Nrf2-dependent [91]. The importance of Nrf2 in suppressing inflammaory cytokines was also studed in neutrophils. LPS-induced elevation of TNF-α, IL-6, and monocyte chemotactic protein-1 (MCP-1), along with increased ROS levels, is more pronounced in Nrf2 knockdown peritoneal neutrophils [63]. Pretreatment of peritoneal neutrophils with the Nrf2 activator CDDO-Imidazole (CDDO-Im) induces Nrf2-dependent antioxidant genes and attenuates changes in proinflammatory cytokines and chemokines in the lungs of Nrf2^+/+^ mice, thereby decreasing mortality [63]. Taken together, the data suggest that Nrf2 is a negative regulator of LPS-TLR4 signaling during innate immune responses by neutrophils and macrophages. In addition, the Nrf2 activator CDDO-trifluoroethyl-amide (CDDO-TFEA) suppresses disease in a murine model of multiple sclerosis by inhibiting Th1 and Th17 mRNA and cytokine production. Importantly, CDDO-TFEA inhibits lymphocyte proliferation and modulates expression of inducible nitric oxide synthase (iNOS) protein in the central nervous system of affected mice [84].

Cyclooxygenases (COX), especially COX-2, are critical inflammatory enzymes that catalyze formation of prostaglandins and thromboxane [92,93]. COX-2 is expressed mainly by macrophages and is upregulated in response to growth factors or inflammatory stimuli such as LPS. The role of iNOS in immune cells has also been established; iNOS is expressed by T cells, macrophages, and mature dendritic cells, and is regarded as a proinflammatory marker [94]. The benificial role of Nrf2 with respect to COX and iNOS signaling is evidenced by aberrant upregulation of COX-2, iNOS, and proinflammatory cytokines, as well as aggravation of non-alcoholic steatohepatitis, in Nrf2 knockout mice [95]. Likewise, a significant increase in oxidative stress, and in expression of COX-2, iNOS, IL-6, and TNF-α, is observed in Nrf2-deficient mice after challenge with proinflammatory stimuli [96]. In addition, pretreatment of primary peritoneal macrophages with sulforaphane potently inhibits LPS-stimulated expression of TNF-α, IL-1β, COX-2, and iNOS mRNA and protein. As expected, these anti-inflammatory effects are attenuated in Nrf2^−/−^ primary peritoneal macrophages, suggesting that this anti-inflammatory activity is mediated in an Nrf2-dependent manner [97]. This notion is further corroborated by evidence showing that the Nrf2 activator Tecfidera inhibits expression of IL-6 and IL-1 genes in experimental models of multiple sclerosis and other autoimmune diseases [98,99]. In an in vitro model of brain inflammation, dimethylfumarate inhibits microglial and astrocytic inflammation by suppressing synthesis of nitric oxide, IL-1β, TNF-α, and IL-6 [100].

Protease and antiprotease levels, as well as oxidative stress, are also thought to be important aspects of inflammation regulated by Nrf2. Matrix metalloproteinases (MMPs) are zinc-dependent proteolytic enzymes responsible for degradation of extracellular matrix components. The Nrf2 activator betulin reduces expression of MMP-13 in IL-1β-induced OA via the AKT/Nrf2/HO-1/NF-κB axis [101]. Likewise, the calcium channel blocker nicardipine, approved by the U.S. Food and Drug Administration (FDA), inhibits metastasis of breast cancer cells via the Nrf2/HO-1 axis and by affecting expression of MMP-9 [102]. In addition, the Nrf2/HO-1 axis inhibits expression of MMP-9 by macrophages, and that of MMP-7 by human intestinal epithelial cells; such inhibition is beneficial for the treatment of inflammatory bowel disease [62,64].

Another study suggests that Nrf2 deficiency induces activation of MAPKs such as JNK, ERK, and p38, in addition to c-Fos, during osteoclast differentiation elicited by proinflammatory stimuli; this suggests that Nrf2 activation might be beneficial with respect to treatment of inflammatory bone diseases [103]. Nrf2/HO-1 signaling in LPS-stimulated macrophages inhibits expression of MMP-9 and iNOS [104], as well as expression of IL-8, ICAM-1, COX-2, and MMP-7 in TNF-α-stimulated human intestinal epithelial HT-29 cells and colonic mucosal tissue [105]. The Nrf2 activator curculigoside increases expression and nucleur translocation of Nrf2, and activities regulatory cytoprotective enzymes via the Nrf2/NF-κB signaling pathway, in RAW264.7 cells. MMP-9 expression is also reduced by the activator [106].

### 3.2. Obesity and Metabolic Syndrome

Obesity is a globally prevalent disease characterized by excessive accumulation of adipose tissue. In the past decade, the role of Nrf2 in the pathogenesis of obesity has been highlighted, as explained in a number of elegant reviews [52,107,108]. To date, the findings regarding the impact of Nrf2 on obesity have been rather controversial. Nrf2^−/−^ mouse embryonic fibroblasts (MEFs) exhibit enhanced adipogenesis upon stimulation, whereas Keap1^−/−^ MEFs, which exhibit higher Nrf2 signaling, show delayed differentiation. When dominant-positive Nrf2 is introduced into Nrf2^−/−^ MEFs, adipocyte differentiation recovers, suggesting a role for Nrf2 in prevention of adipogenesis [109]. By contrast, Nrf2 deficiency in primary cultured mouse preadipocytes and 3T3-L1 cells blocks adipogenic differentiation induced by hormonal cocktails. Importantly, Nrf2 deficiency in 3T3-L1 cells inhibits induction of CCAAT/enhancer-binding protein β, one of the master regulators of adipogenesis; a chromatin immunoprecipitation assay revealed that Nrf2 regulates transcription from the CCAAT/enhancer-binding protein β gene promoter [110]. Furthermore, in MEFs, 3T3-L1 cells, and even in human subcutaneous preadipocytes, selective deficiency of Nrf2 impairs adipocyte differentiation and reduces expression of peroxisome proliferator-activated receptor γ [111]. 

The results of animal studies investigating the impact of Nrf2 modulation on obesity are also conflicting. The Nrf2 activator CDDO-Imidazolide (CDDO-Im) prevents obesity in a murine model of high-fat, diet-induced obesity. Because suppression of lipogenesis by CDDO-Im is inhibited significantly in Nrf2-deficient mice, the effect of CDDO-Im is thought to be Nrf2-dependent [112]. When compared with wild-type mice, hypomorphic Keap1 allele mice (another model of Nrf2 activation) gain less weight, show ameliorated glucose tolerance, and develop less hepatic steatosis. Mechanistically, this is associated with enhanced hepatic AMP-activated protein kinase activity and increased energy expenditure [113]. In agreement with this result, the pharmacological Nrf2 activator oltipraz prevents the detrimental effects of diet-induced obesity on glucose disposal, body weight, and fat gain in mice. The authors observed that levels of nuclear Nrf2 in adipose tissue fell after feeding mice a chronic high-fat diet. Adipose tissue heme oxygenase-1 (HO-1) and superoxide dismutase levels also fell, whereas increased protein oxidation and recruitment of macrophage marker F4/80 in adipose tissue were observed in mice fed a high-fat diet. Administration of oltipraz reversed these detrimental metabolic cues, suggesting that Nrf2 activation is beneficial for preventing obesity [114]. 

Loss-of-function studies also support a beneficial role for Nrf2 in obesity and insulin resistance. Global, as well as adipose-specific, ablation of Nrf2 in *ob/ob* mice results in severe metabolic syndrome [115]. A more recent study generated an adipocyte-specific and hepatocyte-specific Nrf2-deficient mouse model and found that adipocyte-specific Nrf2-deficient mice show impaired glucose tolerance, higher fasting glucose levels, and higher levels of cholesterol and non-esterified fatty acids than control mice, although there were no differences in body weight and energy expenditure. Unlike adipocyte-specific Nrf2-deficient mice, the phenotype of hepatocyte-specific Nrf2-deficient mice is not distinct [116]. Furthermore, an ovariectomized menopausal mouse model shows that Nrf2 KO gain more bodyweight than control mice [117]. In humans, it is noteworthy that obese patients undergoing Roux-en-Y gastric bypass show elevated expression of Nrf2 in adipose tissue after surgery [118]. Taken together, these results suggest that Nrf2 in adipose tissue has a beneficial effect on obesity-related insulin resistance.

However, animal studies of the protective effects of Nrf2 against obesity and systemic insulin resistance are inconsistent, in line with the aforementioned in vitro data. For example, knocking out Nrf2 in diet-induced obesity model mice reduces adipose tissue weight and adipocyte size, thereby protecting against weight gain and obesity [111]. Nrf2 KO mice fed a high-fat diet for 6 months are also partly protected from obesity and insulin resistance. Interestingly, plasma FGF21, and hepatic and adipose tissue Fgf21 mRNA levels in Nrf2-deficient mice are higher than those in wild-type mice; given the beneficial role of FGF21 on metabolic syndrome and obesity, it is postulated that Nrf2 deficiency protects from an obesity-induced metabolic syndrome phenotype, in part by inducing production of FGF21 [119]. Indeed, even after a 12-week high-fat diet, Nrf2 deficiency among Nrf2 deficient mice, wild-type, and Keap1 knockdown mice upregulates hepatic FGF21 and improves glucose tolerance, although body weight change are negligible [120]. 

Xu et al. reported reduced epididymal fat mass and body weight in Keap1 knockdown *ob/ob* mice, which show increased Nrf2 activity. At the same time, however, insulin resistance and glucose intolerance, as well as hepatic steatosis, were aggravated. In vitro experiments using MEFs reveal that both sulforaphane and Keap1 knockdown effectively block adipocyte differentiation [121]. Given that a deficiency in adipose tissue accumulation is related to ectopic fat accumulation, reduced adipose tissue expansion could explain the resulting systemic insulin resistance observed in this study. 

Recent studies suggest that Nrf2 induction might be responsible for adipose tissue browning, adding further complexity to the role of Nrf2 in obesity and metabolic function. Nrf2 induces Ucp1 expression in adipocytes in response to β-3 adrenergic receptor stimulation by activating its promoter [122]. The same researchers also found that sesamol, found in sesame oil, induces Ucp1 and ameliorates obesity in an Nrf2-dependent manner [123].

In summary, although recent evidence suggests that Nrf2 activators might play a role in the treatment of obesity and other related disorders, mechanistic interplay between Nrf2, adipogenesis, and obesity warrants further investigation.

### 3.3. Inflammatory and Autoimmune Diseases

#### 3.3.1. Inflammatory Bowel Disease

Inflammatory bowel disease (IBD) represents a group of intestinal disorders that cause prolonged inflammation of the digestive tract. The inflammation is chronic, with periods of remission and relapse; Crohn’s disease and ulcerative colitis are the two subtypes of IBD. Although 5-aminosalicylic acids such as mesalazine, as well as steroids and antagonists of TNF-α or other inflammatory cytokines, are approved treatments for IBD [124,125,126,127,128], the pathogenesis of the disease is not fully understood.

Disrupted intestinal permeability is the hallmark of IBD. Physiologically, the intestinal mucosa functions as a physical and immunological barrier against various insults. The mucosal barrier comprises an outer mucus layer, intestinal epithelial cells (IECs), and the inner lamina propria, in which innate and adaptive immune cells reside [129]. IECs express a variety of innate immune receptors, such as Toll-like receptors (TLRs), which detect microbes and endogenous danger signals. Intestinal intraepithelial lymphocytes reside between IECs and participate in formation of the intestinal mucosal barrier. Upon pathogen invasion, deregulation of mucosal immunity, or damage to intestinal intraepithelial lymphocytes, intestinal homeostasis is disrupted and inflammation is exacerbated [130]. These phenomena are linked to excessive free radical production and mucosal inflammation [131,132,133,134]. A number of studies show that oxidative stress is excessive in IBD patients. This stress is caused by various factors, including autoimmune abnormalities, changes in the microbiota, and diet. In this regard, ameliortion of oxidative stress by Nrf2 activation could be an ideal mechanism through which to treat IBD [133,135]. Recently, very elegant reviews by Piotroswka et al., Wen et al., and Yanaka have summarized the beneficial role of Nrf2 activation in IBD [135,136,137]. 

Nrf2 activation enhances intestinal barrier integrity in various leaky gut models. In a uremia-induced colonic inflammation model, the pharmacologic Nrf2 activator dh404 restores claudin-1, occludin, and zona occludens protein levels, which are required for optimal barrier function [138]. In line with this, activation of the ERK/Nrf2/HO-1 signaling cascade in a traumatic brain injury model enhances mitophagy, thereby upregulating expression of occludin and zona occludens proteins in the intestinal epithelial layer [139]. In addition, the Nrf2 activator procyanidin B22 protects against oxidative injury in colonic cells and upregulates expression of the antioxidant glutathione S-transferase P1 [140]. Activation of Nrf2 by biogenic nanoselenium protects against epithelial barrier injury; however, its effect is impaired by Nrf2 knockdown, confirming that its activity is Nrf2-dependent [141]. Indeed, Nrf2 binds to the promoter region of claudin-4, thereby increasing its expression [142]; claudin-4 is also downregulated in Nrf2-deficient esophageal epithelial cells [143]. 

Nrf2 activity appears to protect against oxidative stress in the context of IBD. Two distinct models of prohibitin B transgenic mice (a salmonella typhimurium challenge model and a dextran sodium sulfate (DSS)-induced colitis model) exhibit decreased oxidative stress and colitis, predominantly via enhancement of Nrf2 expression in IECs under oxidative conditions [144]. In accordance with this finding, expression of Nrf2, mitogen-activated protein kinase phosphatase 1 (Mkp-1) and HO-1 are increased in colonic tissues of patients with IBD, and in those of DSS-challenged mice [145]. In this study, the authors found that Mkp-1^−/−^ mice are more susceptible to DSS-induced colitis, and that Mkp-1 increases Nrf2 stability [145]. Another experimental model of murine colitis induced by DSS shows that Nrf2-deficient mice have substantially higher levels of lipid peroxidation in the colon than WT mice [146,147]. Indeed, induction of Nrf2 mRNA expression in response to inflammation in IBD colonic tissue highlights the potential cytoprotective role of Nrf2 against oxidative stress [148].

Beyond directly enhancing intestinal epithelial function, immune cell-dependent modulation of IBD activity has also been highlighted. Dehydroepiandrosterone alleviates experimental colitis by inhibiting antiviral immune responses mediated by the NLR family pyrin domain containing 3 (NLRP3) inflammasome in both LPS-stimulated colon epithelial cells and macrophages [149]; these data are consistent with previous observations by another group [150]. By contrast, Nrf2-deficient cells show upregulated expression of cleaved caspase-1, which is attributed to increased transcription of NLRP3 caused by excess ROS [150]. A very recent study shows that expression of Nrf2 and its downstream target HO-1 is increased in BMDMs exposed to apoptotic colonic epithelial cells. Administration of the HO-1 inhibitor zinc protoporphyrin IX blunts resolution of DSS-induced intestinal inflammation, as well as M2 macrophage polarization, suggesting a beneficial role for the Nrf2–HO-1 axis in attenuating colitis via immune function control [151]. The immunomodulatory actions of HO-1 towards M2 polarization have been described by Naito et al. [152].

The role of a number of Nrf2 agonist and antagonists has been studied in IBD models. Ex vivo treatment with the Nrf2 activator CDDO-Im inhibits IL-17 secretion by splenocytes and colonic strips, suggesting that Nrf2 activation blockd T cell hyperactivity, an important pathogenic mechanism in IBD. In a DSS-induced colitis model, CDDO-Im improves altered colonic histology and regulates cytokine (IL-6 and IL-17) expression [153]. Accumulating evidence also shows a beneficial role for DMF in experimental colitis [51,154,155]; this may be due to activation of Nrf2, which results in decreased mitochondrial ROS generation and mitochondrial DNA release, followed by suppression of NLRP3 inflammasome activation [154]. In addition, GB1a, the main active component of Garcinia Kola nuts, attenuates experimental colitis by improving oxidative stress in human colonic epithelial cells via the GB1a-activated Nrf2 antioxidant signaling pathway. This suppresses nuclear translocation of NF-κB, reduces oxidative stress, and reduces expression of inflammatory genes induced by TNF-α [156]. Maresin 1, another Nrf2 activator, attenuates experimental colitis by reducing activation of TLR4/NF-κB. Importantly, ML385, an inhibitor of Nrf2, reverses the protective effects of maresin 1 markedly, indicating that the beneficial effects are Nrf2-dependent [157]. In addition, Nrf2-induced miR-23a-27a-24-2 inhibits expression of Bach1 by binding to the 3’UTR and reducing Bach1-mediated suppression of HO-1 [158]. The Nrf2 activator quercetin reduces diquat-induced oxidative damage to porcine enterocytes, an effect abolished by the Nrf2 inhibitor all-trans-retinoic acid [159]. Finally, imperatorin [160] or olmesartan [161] ameliorate DSS- or TNBS-induced colitis by enhancing the Nrf2-related antioxidant pathway. Taken together, the evidence suggests that Nrf2 activation ameliorates intestinal inflammation, an effect likely mediated by its antioxidant effects and suppression of inflammatory pathways. In addition, the immunomodulatory function of Nrf2 seems critical for blocking inflammation.

#### 3.3.2. Systemic Lupus Erythematous

Systemic lupus erythematous (SLE) is an autoimmune disease that involves inflammation of multiple organs, eventually leading to serious life-threatening complications [158].

The cause of autoimmunity is multifactorial, but interplay between various immune cells is critical [2]. Aberrant IFN signaling, activation of the NLRP3 inflammasome, as well as B cells and Th17 cells, and even hyperactivation of cytototoxic CD8^+^ T cells is noted [162,163,164,165]. Also, T and natural killer cells from SLE patients show higher intracellular ROS levels of than B cells from healthy controls; at the same time, Keap1 and Nrf2 levels are elevated as an antioxidant defense mechanism [166]. Another study of patients with SLE shows that increased circulating plasmacytoid dendritic cells are associated with increased disease activity; in turn, disease activity is positively associated with increased levels of ROS in dendritic cells, due at least partially to reduction of Nrf2 expression. Later in this section, we discuss briefly whether enhanced Nrf2 activity modulates immune cell subsets.

Accumulating evidence from animal studies implicates Nrf2 deficiency in SLE pathogenesis. For example, female Nrf2-deficient mice are prone to developing an autoimmune condition that resembles human SLE [167]. Furthermore, Nrf2 deficiency increases lupus nephritis and Th17 cell numbers in B6/lpr mice [66]. However, similar to IBD, results are contradictory. Another study shows that Nrf2 deficiency improves lupus-prone autoimmune nephritis in a mouse model of lupus, as shown by prolonged lifespan and reduced lymphadenopathy [168].

Several studies indicate that Nrf2 activation has a beneficial effect on pathogenic immune cells. A pristane-induced lupus mouse model shows that the Nrf2 activator CDDO-Im decreases expression of type I IFN receptor and IFN-stimulated genes in macrophages, and alleviates oxidative stress, whereas the Nrf2 inhibitor brusatol has the opposite effect. Likewise, IFN receptor expression in Nrf2-knockout mice is higher than that in controls [169]. The Nrf2 activator SM934, a water-soluble derivative of artemisinin, improves the lifespan of lupus-prone MRL/lpr mice. It also reduces disease activity (including lymphadenopathy), as well as serum levels of anti-nuclear antibodies and cytokines (IL-6, IL-10, and IL-2). Specifically, SM934 increases quiescent B cell numbers and maintains germinal center B cell numbers in the spleen, while at the same time decreasing the numbers of activated B cells and plasma cells. A ex vivo experiment showed that SM934 suppresses TLR-triggered activation and proliferation of B cells, an effect recapitulated in human peripheral blood mononuclear cells [170]. The Nrf2 activator A-1396076 dampens inflammation in an IFN-α-accelerated NZB/W mouse lupus nephritis model by inhibiting antigen-dependent T cell activation [171]. In addition, CDDO-Me reduced severity of lupus disease by attenuating MEK-1/2, ERK, and STAT-3 signaling in CD4^+^ T cells, as well as oxidative stress in B6.Sle1.Sle3 mice or MRL/lpr mice, both of which develop spontaneous lupus [172].

Moreover, Nrf2 activators such as sulforaphane and DMF show anti-inflammatory effects in human renal mesangial cells. DMF ameliorates development of kidney disease in pristane-induced lupus nephritis mice and shows stronger anti-inflammatory and organ-protective effects than glucocorticoids [173]. Likewise, 4-octyl itaconate activates Nrf2 signaling, thereby inhibiting production of proinflammatory cytokines in human macrophages and SLE patient-derived peripheral blood mononuclear cells [90]. Finally, Nrf2 activation by sulphoraphane suppresses pritane-induced lupus nephritis by neutralizing ROS, thereby preserving renal function [174]. The same study shows that Nrf2 KO mice are more susceptible than their wild-type littermates to lupus induced by pristine injection [174].

#### 3.3.3. Rheumatoid Arthritis (RA)

RA is a chronic, autoimmune and inflammatory disease [175]. Pathophysiologically, the disease is characterized by inflammation of the lining of the joints (synovitis), eventually leading to destruction of the cartilage and underlying bone [175]. These effects are mediated by activation and migration of neutrophils, macrophages, and lymphocytes. This results in increased production of proinflammatory mediators such as oxidants, eicosanoids, and cytokines, and triggers hyperproliferation of synovial fibroblasts. In the synovial membrane and adjacent bone marrow, adaptive and innate immune pathways integrate to promote tissue remodeling and joint damage. Positive feedback loops mediated by interactions among leukocytes, synovial fibroblasts, chondrocytes, and osteoclasts, together with the molecular products of tissue/bone damage, drive the chronic phase of RA [175]. 

Oxidative stress plays a detrimental role in the pathophysiology of RA [176]. Indeed, emerging evidence from various disease models shows that activation of Nrf2 improves RA. Resveratrol alleviates oxidative stress and apoptosis in hydrogen peroxide-treated fibroblast-like synoviocytes by activating the Nrf2-Keap1 pathway [177]. In rats, DMF ameliorates complete Freund’s adjuvant-induced arthritis [178] by suppressing oxidative stress markers and inflammatory mediators, and by increasing Nrf2 and HO-1 levels in the involved joints [178].

Recent evidence suggests that not only Th1, but also Th17 cells, which produce IL-17A, 17F, 21, and 22, are involved in RA pathogenesis [179,180,181]. When the Nrf2 activator kurarinone is administered orally to mice with collagen-induced arthritis, levels of proinflammatory cytokines (TNF-α, IL-6, IFN-γ, and IL-17A) fall in both serum and paw tissues. In addition, kurarinone reduces phosphorylation of STAT1 and STAT3, as well as the numbers of Th1 and Th17 cells in lymph nodes, suggesting that kurarinone exerts an anti-inflammatory effect by inhibiting Th1 and Th17 cell differentiation [182].

Furthermore, results from a murine collagen-induced arthritis model show that pharmacological activation of Nrf-2 attenuates joint inflammation and destruction by activating the Ho-1 antioxidant pathway and by suppressing activation of mitogen-activated protein kinase (MAPK) and NF-κB [183]. Liu et al. studied the role of nuclear receptor subfamily 1 group D member 1 (NR1D1), a transcriptional repressor, in an RA model. They showed that NR1D1 activation reduces ROS generation and increases production of Nrf2-associated enzymes in RA fibroblast-like synoviocytes. The NR1D1 agonist SR9009 significantly suppresses synovial hyperplasia, infiltration of inflammatory cells, and destruction of cartilage and bone in mice with CIA. Although this compound was not tested in Nrf2 KO mice, it is very likely that Nrf2 activation plays a pivotal role in NR1D1 activation-mediated improvements in RA [184]. DC32, a dihydroartemisinin derivative, induces Keap1 degradation and activates the Nrf2/HO-1 pathway in CIA mice, as well as inhibiting LPS-induced inflammatatory responses in NIH-3T3 cells. The authors conclude that DC32 significantly suppresses RA via the Nrf2-p62-Keap1 feedback loop by increasing the mRNA and protein levels of Nrf2, and by inducing expression of p62 [185]. In addition, recent studies support the benificial role of Nrf2 in RA [24,186,187]. Taken together, the data suggest that activation of the Nrf2 pathway shows unequivocal preventive or therapeutic effects against RA. Table 1 summarizes the role of Nrf2 in inflammation-related phenotypes.

## 4. Nrf2 Agonism in Diabetes and Its Complications

### 4.1. Diabetes

Diabetes mellitus is a metabolic disorder characterized by chronic hyperglycemia resulting from defects in insulin secretion, insulin sensitivity, or both [188]. The main pathophysiology of both type 1 diabetes and type 2 diabetes is loss of pancreatic β-cell function [189]. Pancreatic β-cells, one of the most metabolically active tissues in the human body, are highly dependent on oxidative phosphorylation for ATP synthesis, especially under high glucose conditions [190]. Generation ROS is a consequence of mitochondrial respiration in response to increased availability of glucose and other substrates. However, expression of antioxidant defense genes is relatively low in β-cells [191], which makes them more vulnerable to damage caused by oxidative stress [192]. Hyperglycemia, particularly in diabetes, induces high levels of oxidative stress in the pancreatic β-cells of affected patients [193]; chronic exposure to high levels of oxidative stress leads to β-cell dysfunction and death [194]. 

In healthy β-cells, acute transient hyperglycemia-induced oxidative stress is regulated by transcription of ARE-driven genes (Figure 4) [195]. Nrf2 is a master regulator of ARE-driven genes in pancreatic β-cells [196,197]. During hyperglycemia, insulin secretion is increased along with expression of detoxifying enzymes and antioxidants [198]. In response, ROS are attenuated by Nrf2 [199] and β-cells are protected from oxidative stress. However, chronic exposure to hyperglycemia leads to accumulation of ROS due to incomplete reduction of oxygen during glucose metabolism [200,201]. In addition, aberrant glucose metabolism generates ROS through activation of protein kinase C, glucose auto-oxidation, generation of excessive superoxide, increased hexosamine metabolism, and increased islet amyloid deposition [202,203,204]. A major factor contributing to pancreatic β-cell dysfunction is oxidative stress-mediated mitochondrial damage [205]. Therefore, chronic exposure to high ROS levels can result in β-cell dysfunction, impaired glucose-induced insulin secretion, and β-cell apoptosis [206].

Significant reduction in expression of Nrf2-related cytoprotective genes, excessive release of ROS, and depletion of antioxidant defense systems occur in the pancreatic β-cells of diabetic murine models (db/db mice) [207]. By contrast, Nrf2 activation in Keap1 hypomorphic knockdown (Keap1^flox/-^) models, and by oral administration of an Nrf2 inducer (RTA-403), improves both insulin secretion and insulin sensitivity [208]. Nrf2 induction in diabetic mice also suppresses gluconeogenesis owing to transcriptional repression of several enzymes, including the gluconeogenic enzyme glucose-6-phosphatase [209]. It is widely accepted that Nrf plays an essential role in diabetes prevention by controlling oxidative stress [192,210]. Various Nrf2 activators have been tested in clinical trials (Table 2). The results show improved insulin sensitivity, lower fasting glucose levels, reduced hepatic glucose production, improved lipid profiles, and reduced expression of inflammatory markers. See Section 5.1 for further discussion about the use of Nrf2 activators to treat diabetes.

### 4.2. Diabetic Complications

Diabetic complications occur in more than half of diabetic patients and are the leading cause of diabetes-related death [211,212]. Macrovascular complications induced by accelerated atherosclerosis increase the risk of myocardial infarction, stroke, and lower limb amputation; microvascular complications, including retinopathy and nephropathy, are the main contributors to adult blindness and renal failure [213]. ROS generation is key to development of diabetic complications [214,215]. ROS generation exceeds the scavenging capacity of the cellular antioxidant system, resulting in inflammation and damage to proteins, lipids, and DNA. This damage exacerbates oxidative stress. In blood vessels, this contributes to generation and aggravation of atherosclerosis via inflammation-induced fibrosis, smooth muscle cell proliferation, tunica media thickening, accumulation of lipid, plaque formation, and calcification [216,217]. Type 1 diabetes (T1D), also known as insulin-dependent diabetes, is considered an autoimmune metabolic disorder. However, among the numerous factors that contribute to T1D complications, such as diabetic nephropathy, retinopathy, polyneuropathy, and cardiovascular disease, hyperglycemia-induced ROS are a key driver secondary complications. Thus, Nrf2 provides cellular protection, ameliorates oxidative stress and inflammation, and delays progression of diabetes-related complications [218].

As mentioned previously, Nrf2 plays a critical role in cellular defense against oxidative stress. Therefore, Nrf2 activators or Keap1 inhibitors are attractive targets for drugs [219]. Sulforaphane (SFN) is a phytochemical that induces Nrf2 activation by modifying cysteine residues (C151) of Keap1 [220]. Several studies in animal models with diabetic complications show that SFN protects against diabetic cardiovascular disease via Nrf2 activation [221,222,223,224]. In advanced glycation end product (AGE)-exposed human umbilical vein endothelial cells and AGE-injected rat aorta, SFN suppresses expression of the genes encoding receptor of AGE (RAGE), monocyte chemoattractant protein-1, intercellular adhesion molecule-1, and vascular cell adhesion molecular-1 in a dose-dependent manner [221]. In Goto-Kakizaki rats, an animal model of non-obese type 2 diabetes, SFN improves endothelial dysfunction in the aorta and mesenteric arteries, thereby decreasing vascular oxidative damage, and AGE and hemoglobin A1c levels [222]. Dimethyl fumarate (DMF) is a synthetic Nrf2 activator that alkylates Keap1 cysteine residues; it is used clinically to treat multiple sclerosis [225]. Studies in a streptozotocin (STZ)-induced, diabetes-associated rat model of vascular complications show that DMF treatment ameliorates hyperglycemia, lowers serum AGE levels, and reduces ROS levels in aortic tissue by modulating the endogenous thioredoxin redox system [226]. In addition to macrovascular complications, Nrf2 activators improve microvascular complications such as diabetic neuropathy and nephropathy [227,228,229,230]. In an STZ-induced diabetic murine model, genetic knockout of Nrf2 is associated with development of diabetic nephropathy. In line with this, administration of Nrf2 activators (SFN or cinnamic aldehyde) reduce albuminuria and minimize pathological alterations of the glomerulus in Nrf2^+/+^, but not in Nrf2^−/−^, mice [231]. Another study using the STZ-induced diabetic neuropathy rat model investigated the effects of an Nrf2-inducer (SFN) on diabetic neuropathy. The results show that SFN administration improves nerve conduction velocity, nerve blood flow, and pain behavior, and reduces levels of the oxidation stress marker malondialdehyde [232]. In db/db mice, SFN ameliorates expression of genes related to Nrf2 in the microvasculature, and that of associated genes such as glutathione synthesis-related enzyme; SFN treatment also increases Nrf2 and dampens ROS signaling markedly via upregulation of the glutathione system [233]. Another diabetic microvascular complication is diabetic retinopathy, which can lead to blindness [234,235]. 

The neuroprotective effects of Nrf2 against diabetic encephalopathy were investigated in the STZ-induced diabetic rat model. Oxidative stress-mediated injury occurred in diabetic rats. However, insulin treatment promoted translocation of Nrf2 to the nucleus and activated exporession of downstream antioxidant proteins [236].

The effect of diabetes on Nrf2 was investigated in the retina of an STZ-induced diabetic rat model. The results show that diabetes reduces the DNA-binding activity of Nrf2; however, glucose-induced impairment of Nrf2 in retinal endothelial cells is prevented by an Nrf2 inducer (tert-butyhydroquinone) and by Keap1 siRNA [237]. This suggests that activation of Nrf2 prevents progression of diabetic retinopathy [238,239]. 

In addition to diabetic complications, decreased Nrf2 levels and activity are reported in studies of aging. Nrf2 plays an essential role in regulating cell senescense [50]. For example, melatonin prevents aging of adipose-derived mesenchymal stem cells by activating Nrf2 and inhibiting ER stress [240]. 

Taken together, evidence suggests that activation of the Nrf2 pathway could be a successful therapeutic target in the treatment of diabetes and related complications [218,241,242].

## 5. Recent Development of Nrf2-Related Drugs and Performance in Clinical Trials

### 5.1. Nrf2 Pharmacological Activators under Clinical Trial

Several Nrf2 pharmacological activators have been, or are currently being, tested in clinical trials for the treatment of various diseases (Table 2) [243]. Because there are so many clinical trials related to Nrf2, we will only summarize those related to diseases mentioned in this article. Trials of chronic inflammatory diseases, IBD, and autoimmune diseases such as SLE are still at the preclinical stage, so there are few data regarding the progress of prospective clinical trials. Bardoxolone methyl (CDDO-Me; RTA-402) is an oleanolic acid-derived, semi-synthetic triterpenoid. CDDO-Me, a strongly electrophilic cyanoenone, binds covalently and reversibly to the sulfhydryl groups of Keap1, thereby triggering structural changes that prevent Nrf2 ubiquitination [38]. This allows activated Nrf2 to translocate to the nucleus and upregulate antioxidant and cytoprotective genes.

In 2019, a phase I Study of RTA-402 was conducted in obese adult patients (NCT04018339). The study started in August 2019 and was completed by May 2020. Researchers investigated changes in body weight and body composition of obese adults after repeated oral administration of RTA-402 once daily for 16 weeks, using placebo as a control. The results have not yet been reported. 

CDDO-Me has long been investigated for its ability to treat chronic kidney disease [244]. Now, it is under phase 3 clinical trial (CARDINAL study; NCT03019185) for the treatment of chronic kidney disease (CKD) caused by Alport syndrome; the data were submitted for approval in the United States in 2021 [245]. In addition, CDDO-Me is in phase III clinical development for the treatment of CKD caused by autosomal dominant polycystic kidney disease (The FALCON study; NCT03918447) and pulmonary arterial hypertension (The RANGER study; NCT03068130). Studies show that the Nrf2 pathway in the lung tissue of individuals infected with severe acute respiratory syndrome coronavirus 2 (SARS-CoV-2) is suppressed, and that induction of Nrf2 by 4-octyl-itaconate and dimethyl fumarate inhibits replication of SARS-CoV-2 [246]. As a result, CDDO-Me is being evaluated in phase II/III clinical trials for the treatment of COVID-19 (The BARCONA study; NCT04494646). Another phase III clinical trial of CDDO-Me was conducted to examine its ability to delay progression to end-stage renal disease in patients with CKD and type 2 diabetes; however, these studies were discontinued in 2012 due to serious adverse events and mortality [247]. In the CDDO-Me group, 96 patients were either hospitalized with heart failure or died from heart failure; these events occurred in only 55 particpants in the placebo group (hazard ratio, 1.83; 95% CI, 1.32–2.55; *p* < 0.001). Nevertheless, new phase II/III studies are underway in patients with diabetic kidney disease, type 2 diabetes, and rare forms of CKD, including IgA nephropathy, focal segmental glomerulosclerosis, and CKD-associated type 1 diabetes [248,249]. 

Similar to CDDO-Me, omaveloxolone (RTA-408), also from the bardoxolone family, is being tested in clinical trials as a treatment for nondiabetic conditions such as liver cirrhosis (NCT03902002), cataracts (NCT02128113), Friedreich’s ataxia (NCT02255435), and mitochondrial myopathy (NCT02255422) [250,251,252].

Other promising candidates belonging to the CDDO family are CDDO-Im (CDDO-Imidazolide; RTA-403; TP-235) and dh-404 (CDDO-dhTFEA; RTAdh-404). These are currently being developed as therapeutic agents for various diseases (e.g., as chemopreventive agents, glioblastoma multiforme therapy, pulmonary emphysema therapy, and antidiabetic, antiobesity, renoprotective, and cardioprotective drugs) [253,254]. These drugs are still at the preclinical stage.

Two synthetic Nrf2 activators have been approved by the FDA. DMF (FDA approval, brand name Tecfidera^®^) was approved for the treatment of relapsing-remitting multiple sclerosis [225] and psoriasis, and ursodiol (brand names Actigall^®^ and Urso^®^) was approved for the treatment of primary biliary cirrhosis [58]. DMF is the methyl ester of fumaric acid, a biologically active electrophile. DMF inhibits Nrf2 degradation via Keap1 succinylation. The released Nrf2 promote transcription of antioxidant and cytoprotective genes [255]. Phase III clinical trials are ongoing for the oral treatment of pediatric patients with relapsing-remitting multiple sclerosis (NCT03870763) [256]. A phase I trial is ongoing for the treatment of pulmonary arterial hypertension associated with scleroderma (NCT02981082) [257]. 

Ursodiol (FDA approval, brand names Actigall^®^ and Urso Forte^®^), a bile acid originally isolated from the Chinese black bear, was launched in 1980 for the treatment of primary biliary cirrhosis and gallstones [258]. In July 2004, the FDA approved ursodiol for the treatment of primary biliary cirrhosis. Although the mode of action has not been fully elucidated, several research groups suggest that upregulation of Nrf2 is the main mechanism [259,260,261]. Phase I clinical trials are also underway for the treatment of Huntington’s disease. In the EU and U.S., the compound was assigned orphan drug status for the treatment of Niemann–Pick disease in 2017 and 2018, respectively [262]. 

AJ-101 (formerly ALZ-001), an Nrf2 activator, is in phase II clinical trials for the treatment of acne vulgaris, and in phase I clinical trials for the treatment of cutaneous polycystic ovary syndrome [263].

Oltipraz, an Nrf2 agonist, has been in phase II clinical trials for the treatment of liver fibrosis, cirrhosis, and non-alcoholic fatty liver disease [264,265]. 

An increasing number of natural Nrf2 modulators have been discovered and are of great interest [266]. Among the natural Nrf2 agonists, SFN has been actively studied, and a phase II clinical trial for the treatment of patients with type 2 diabetes is ongoing (NCT02801448). Sulforadex (SFX-01) is an Nrf2 activator composed of SFN and alpha-cyclodextrin, which is undergoing phase II/III clinical trials for the treatment or prevention of acute respiratory distress syndrome related to COVID-19. It is also being tested in a phase II clinical trial as a candidate for the treatment of subarachnoid hemorrhage (ClinicalTrials.gov Identifier: NCT02614742) [267]. In 2016, SFX-01 was designated as an orphan drug in the U.S. for the treatment of subarachnoid hemorrhage [267]. In addition to SFN, clinical trials show that the natural Nrf2 agonist curcumin can lower blood sugar in patients with type 2 diabetes; resveratrol and quercetin are also in phase III clinical trials for type 2 diabetes [268].

Several clinical trials using synthetic or natural Nrf2 agonists are underway for the treatment of diabetes, diabetic complications, non-alcoholic steatohepatitis, and dyslipidemia. These are summarized in Table 2 and Figure 5.

## 6. Conclusions

Overproduction of ROS and oxidative stress due to an imbalance in the levels of oxidants and antioxidants plays a crucial role in the pathogenesis of many diseases. In metabolically active cells, the Nrf2 pathway orchestrates cellular redox homeostasis by regulating the antioxidant signaling axis and neutralizing various oxidative agents. Consequently, the Keap1–Nrf2–ARE pathway is a promising target for regulation of numerous diseases associated with oxidative stress and inflammation.

As shown in Figure 1, evidence argues against dissociation of the Keap1/Nrf2 complex upon oxidation of Keap1 cysteines [38]. Rather, under oxidative stress, p62 accumulation disrupts the DLG-Keap1 interaction and releases Nrf2 via conformational changes. Therefore, newly synthesized Nrf2 is no longer ubiquitinated and degraded. Rather, it accumulates rapidly and translocates to the nucleus. Based on this new molecular mechanism, new small molecules could be designed to overcome the limitations of traditional Nrf2 activators that act through cystine modification of Keap1.

In addition, the relative contributions of these two distinct molecular mechanisms to inhibition of Keap1 function under oxidative stress remain unclear. Further studies are needed to clarify this issue. 

Despite promising results in preclinical trials, it is noteworthy that the majority of compounds examined in animal studies are phytochemicals (derived from natural plant sources). To achieve a successful bench-to-bedside strategy, improvements in the biological activity and targeting of these compounds are required. 

As discussed above, an increasing number of clinical trials have verified the beneficial effects of Nrf2 on metabolic/inflammatory diseases. At the moment, many Nrf2 activators with diverse chemotypes are at the clinical trial stage; these include DMF, SFN, CDDO-Me, and their derivatives.

An interesting point about the pleotropic effects of Nrf2 agonists is how these compounds sense cysteines in various cellular targets. Modulatory effect mediated via sensing of cysteine, not only cysteines in Keap1, but also cysteine sulfhydryl groups in glutathione (GSH) and proteins, could explain the pleiotropic effects of Nrf2 agonists such as sulforaphane [269]. Further research needs to be carried out before these agents can be used in the clinic. More comprehensive and in-depth controlled studies are needed to explore the role of Nrf2 in the pathogenesis of metabolic/inflammatory diseases, along with safety outcomes, before therapeutic targeting of Nrf2 becomes a clinical reality.

## Figures and Tables

**Figure 1 ijms-23-02846-f001:**
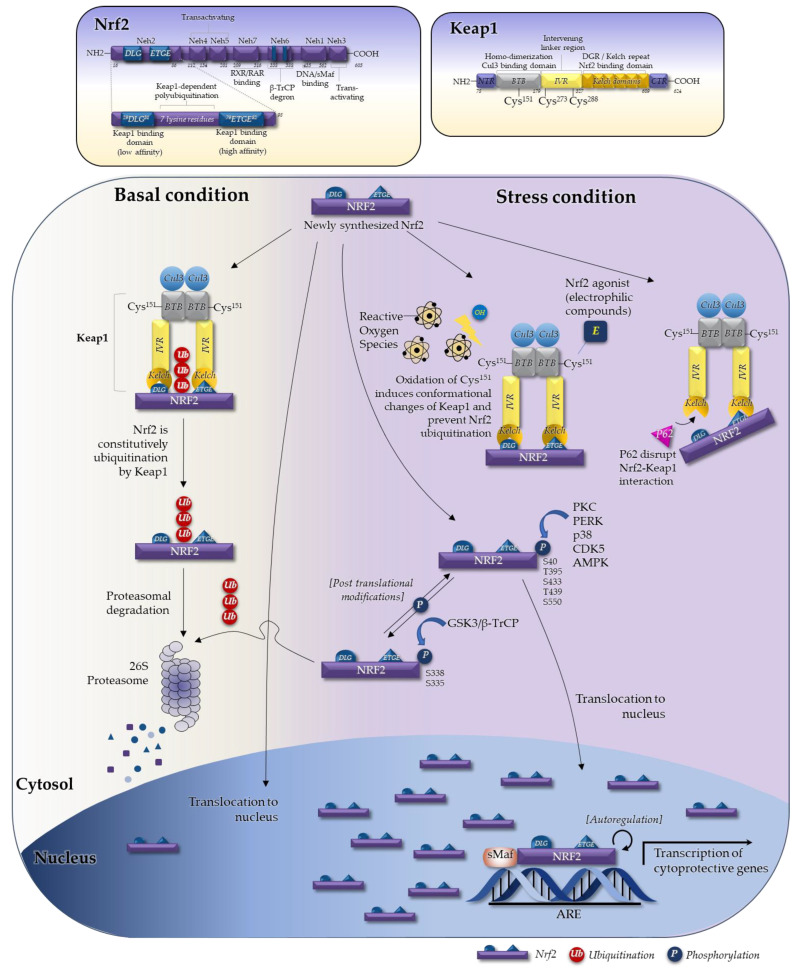
The Nrf2–Keap1 UPS system is the major pathway of Nrf2 regulation. Nrf2 activity is regulated by various mechanisms such as ubiquitination, post-translational modifications, and autoregulation. In the Keap1–Nrf2 UPS system, the Keap1 homodimer binds to the ETGE and DLG motifs of the Neh2 domain of Nrf2. Under basal conditions, Keap1 binds to Nrf2, is subsequently ubiquitinated by the Keap1–Cul3 complex, and is then degraded by the 26S proteasome. Under oxidative stress conditions, oxidation of the cysteine residues of Keap1 causes conformational changes in Keap1 and prevents Nrf2 ubiquitination, thereby allowing entry of newly synthesized Nrf2 to the nucleus. Majority of the Nrf2 agonists, which are electrophilic, covalently bind to Keap1-C^151^ cysteine residues, thereby cause conformational changes of Keap1 and prevent Nrf2 ubiquitination. P62, a ubiquitin-binding protein, competes with Nrf2 for binding to Keap1. Accumulation of P62 disrupts the DLG-Keap1 interaction and releases Nrf2. In addition, phosphorylation of Nrf2 dissociates from Keap1 and translocates to the nucleus. Through heterodimerization with sMAF proteins, Nrf2 binds to cis-acting consensus DNA sequences (referred to as AREs). The Nrf2–sMAF complex activates transcription of cytoprotective genes. Nrf2 activates expression of its own gene (a process called autoregulation), leading to increased production of Nrf2 protein. Abbreviations: AMPK, AMP-activated protein kinase; ARE, antioxidant responsive element; β-TrCP, beta-transducin repeats-containing protein; Cul3, cullin3; CDK5, cyclin dependent kinase 5; GSK3, glycogen synthase kinase-3; Keap1, Kelch-like ECH-associated protein 1; BTB, Broad complex, Tramtrack and Bric-à-Brac; IVR, Intervening linker region; Neh, Nrf2-erythroid cell-derived protein with CNC homology; Nrf2, nuclear factor-erythroid-derived 2-related factor 2; PERK, PKR-like ER kinase; PKC, protein kinase C; UPS, ubiquitin proteasome degradation system; sMAF, small musculoaponeurotic fibrosarcoma.

**Figure 2 ijms-23-02846-f002:**
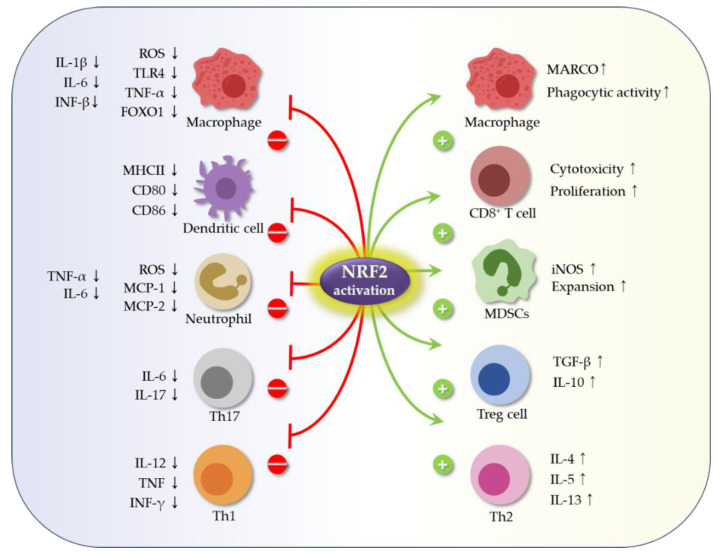
Nrf2 regulates immune cells and inflammation. Activation of Nrf2 can alter the differentiation, expansion, and survival of immune cells, as well as release of cytokines. Increased expression of Nrf2 by macrophages suppresses expression of proinflammatory genes [61]. Nrf2 upregulates MARCO, a scavenger receptor for bacteria, which increases binding and phagocytosis, thereby enhancing bacterial clearance [68]. Nrf2 also upregulates stimulation of antigen-specific CD8+ T cells [69]. Nrf2 activation suppresses the inflammatory response of dendritic cells [70]. In neutrophils, Nrf2 decreases ROS production and expression of TNF-α, IL-6, MCP-1, and MIP-2. Nrf2 activation impairs Th1-driven responses and biases them towards Th2 differentiation [63]. Nrf2-mediated antioxidant defenses induce expansion and survival of Treg cells [71]. Nrf2 deficiency increases oxidative damage, which exacerbates differentiation of Th17 cells [66]. Nrf2 activation in MDSCs leads to expansion of inhibitory MDSCs [72]. Abbreviations: IL-6, interleukin-6; INF-γ, interferon gamma; iNOS, inducible nitric oxide synthase; MARCO, macrophage receptor with collagenous structure; MCP-1, monocyte chemoattractant protein-1; MDSCs, myeloid-derived suppressor cells; MHCII, major histocompatibility complex II; MIP-2, macrophage inflammatory protein-2; Nrf2, nuclear factor-erythroid-derived 2-related factor 2; ROS, reactive oxygen species; TGF-β, transforming growth factor-beta; TLR4, toll-like receptor 4; TNF-α, tumor necrosis factor-alpha.

**Figure 3 ijms-23-02846-f003:**
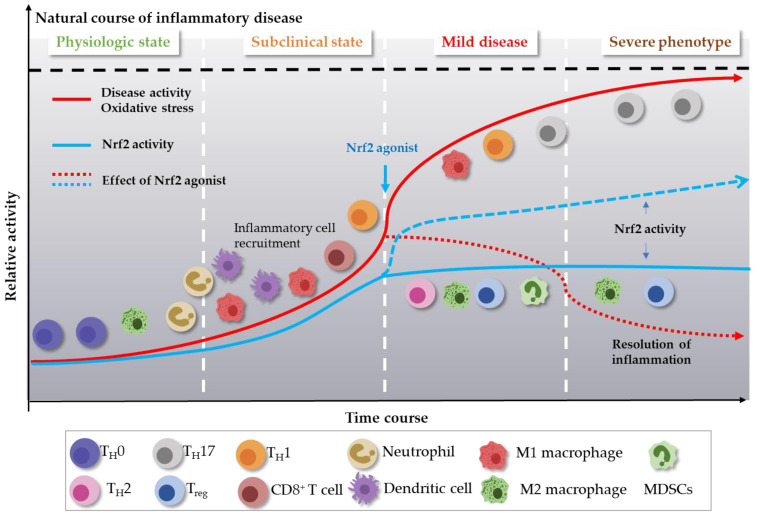
Links between immune cells, inflammatory disease activity, and levels of oxidative stress. Exposure of the immune system to oxidative stress increases inflammatory disease activity. Oxidative stress is sensed by the Nrf2 system, which induces intracellular signaling pathways in immune cells, thereby triggering an immediate innate immune response, followed by differentiation of adaptive immune cells. Ideally, an effective compensatory response is elicited by increased expression of Nrf2, which restores homeostasis. Severe oxidative stress that can be more than physiologically compensated by the Nrf2 response leads to an increase and uncompensated response in the form of excessive inflammation; for example, increased expression of Th17. This leads to aggravation of inflammatory disease activity. Thus, administration of Nrf2 agonist can play an important role in regulating immune homeostasis.

**Figure 4 ijms-23-02846-f004:**
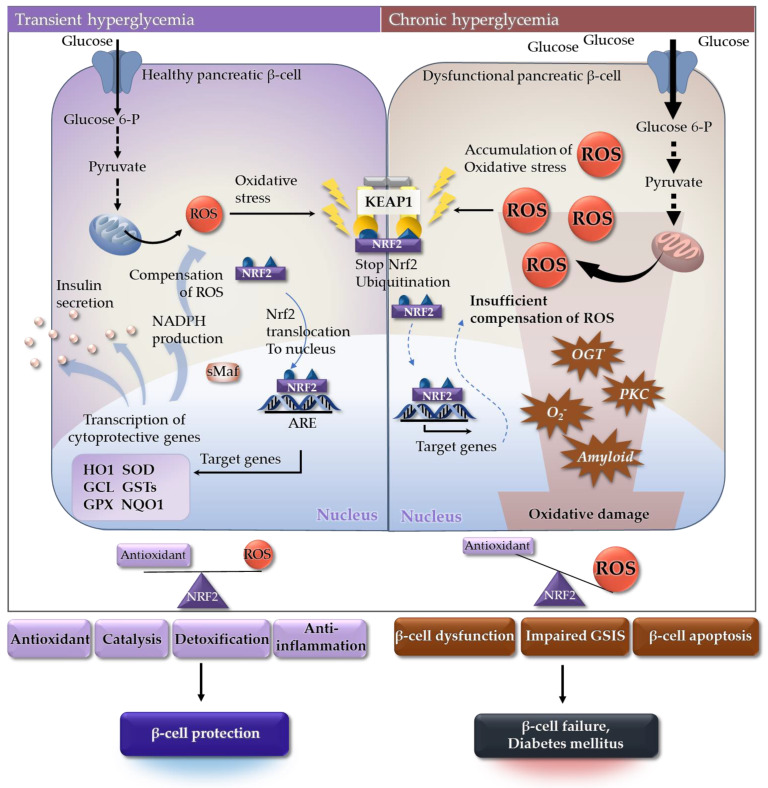
Oxidative stress-centric pathophysiology of β-cell dysfunction. (**Left**) In healthy β-cells, transient hyperglycemia-induced oxidative stress is regulated by Nrf2-mediated transcription of ARE-containing genes. Expression of detoxifying enzymes and antioxidants, as well as insulin secretion, is elevated. During this process, ROS are scavenged by Nrf2-related antioxidant proteins in β-cells. (**Right**) Chronic exposure to hyperglycemia leads to accumulation of ROS. Most ROS originate from mitochondria due to incomplete reduction of oxygen during oxidative phosphorylation. However, aberrant glucose metabolism generates more ROS through altered gene expression and additional pathways, including the formation of AGEs, activation of PKC, increased hexosamine metabolism, and increased islet amyloid deposition. Insufficient compensation for ROS results in β-cell dysfunction, impaired GSIS, and β-cell apoptosis. Abbreviations: AGEs, advanced glycation end-products; ARE, antioxidant responsive element; GCL, glutathione cysteine ligase; Glucose-6-P, glucose-6-phosphate; GPX, glutathione peroxidase; GSIS, glucose-induced insulin secretion; GSTs, glutathione S-transferases; HO1, heme oxygenase-1; KEAP1, Kelch-like ECH-associated protein 1; NADPH, nicotinamide adenine dinucleotide phosphate; NQO1, NADPH quinone dehydrogenase 1; Nrf2, nuclear factor-erythroid-derived 2-related factor 2; OGT, O-GlcNAc transferase; PKC, protein kinase C; ROS, reactive oxygen species; sMAf, small musculoaponeurotic fibrosarcoma; SOD, superoxide dismutase.

**Figure 5 ijms-23-02846-f005:**
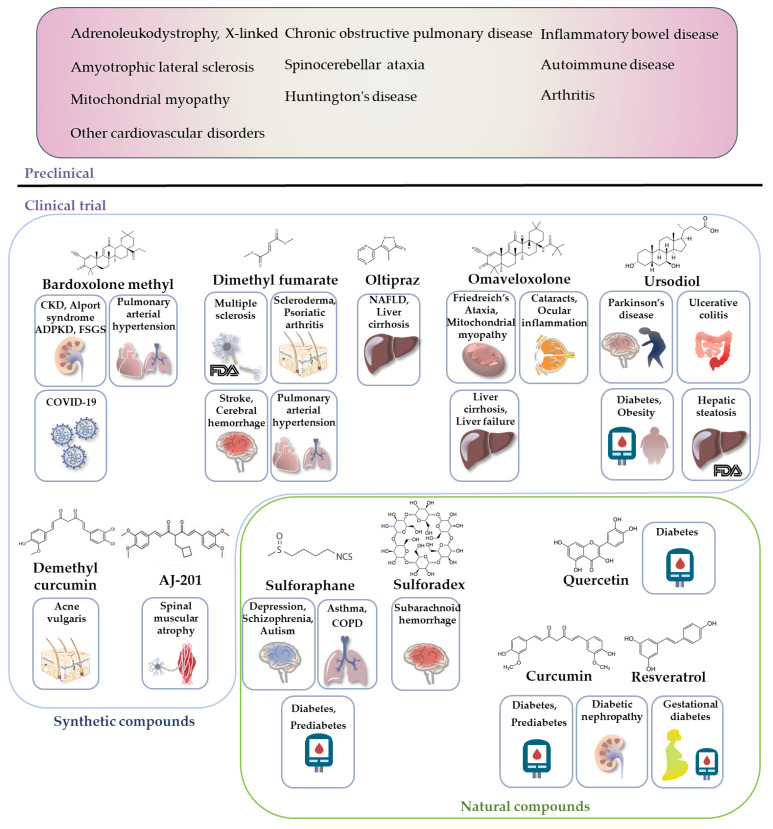
Preclinical and clinical trials of Nrf2 activators in metabolic/inflammatory diseases. Nrf2 activators are thought to reduce ROS, thereby preventing many chronic metabolic/inflammatory diseases caused by oxidative tissue damage. Preclinical and clinical studies provide evidence that Nrf2 activators are therapeutic agents for metabolic/inflammatory diseases such as adrenoleukodystrophy, inflammatory bowel disease, amyotrophic lateral sclerosis, spinocerebellar ataxia, autoimmune diseases, Huntington’s disease, arthritis, CKD, diabetes, and cardiovascular disorders. Abbreviations: ADPKD, autosomal dominant polycystic kidney disease; CKD, chronic kidney disease; CVD, cardiovascular disease; COVID-19, Coronavirus disease 2019; NAFLD, non-alcoholic fatty liver disease; COPD, chronic obstructive pulmonary disease.

**Table 1 ijms-23-02846-t001:** The role of Nrf2 in major inflammatory diseases (animal studies).

Disease	Animal Model	Nrf2 Gain-of Function (Including Nrf2 Agonist) or Loss-of-Function	Phenotype	Refs.
Obesity	C57BL/6J mice WT and Nrf2-disrupted; high fat diet (HFD)	Nrf2 loss-of-function, Nrf2 agonist (CDDO-Im)	Improvement of obesity and suppression of lipogenesis by CDDO-Im. This beneficial is lost in Nrf2-deficient mice	[112]
	C57BL/6J mice WT and Keap1-hypo; HFD	Nrf2 gain-of function (Hypomorphic Keap1 allele)	Hypomorphic Keap1 allele mice (model of Nrf2 activation) gain less weight, show ameliorated glucose tolerance, and develop less hepatic steatosis. Keap1-hypo livers exhibit activated AMPK signaling	[113]
	C57BL/6J mice; HFD	Nrf2 agonist (Otipraz)	Administration of Nrf2 activator reverses the detrimental effects of HFD-induced obesity.	[114]
	C57BL/6J *ob/ob* mice	Nrf2 loss-of-function (Global and adipocyte-specific)	Global, as well as adipose-specific, ablation of Nrf2 in *ob/ob* mice results in severe metabolic syndrome	[115]
	C57BL/6J HFD	Nrf2 loss-of-function (Adipocyte-specific Nrf2^−/−^)	Adipocyte-specific Nrf2^−/−^ impaires glucose tolerance, higher fasting glucose levels, and higher levels of cholesterol and non-esterified fatty acids.	[116]
	*ob/ob* mice	Nrf2 gain-of function (Keap1^−/−^)	Reduces epididymal fat mass and body weight	[121]
Inflammatory Bowel Disease	salmonella typhimurium challenge model, dextran sodium sulfate (DSS)-induced colitis model	Nrf2 gain-of function by prohibitin B overexpression (transgenic mice)	Prohibitin B transenic mice exhibit decreases oxidative stress and improved colitis	[144]
	DSS-induced colitis	Nrf2 loss-of-function (Mitogen-activated protein kinase phosphatase 1 KO; Mkp-1^−/−^)	Mkp-1^−/−^ mice are more susceptible to DSS-induced colitis	[145]
	DSS-induced colitis	Nrf2-deficient mice	Increased inflammation and mucosal damage	[146]
	DSS-induced colitis	Dehydroepiandrosterone (DHEA); acivates Nrf2 via G protein-coupled receptor 30 (GPR30)-dependent pathway	DHEA inhibits intestinal inflammation and improves barrier function in DSS-induced colitis model	[149]
	DSS-induced colitis	Nrf2 inhibition by treating HO-1 inhibitor zinc protoporphyrin IX (ZnPP)	Administration of ZnPP blunts the resolution of DSS-induced intestinal inflammation and expression of the proresolving M2 macrophage marker CD206	[151]
	DSS-induced colitis	Nrf2 activation by CDDO-Im	Administration of CDDO-Im improves the altered colonic histology, and cytokine	[153]
	DSS-induced colitis	Nrf2 activation by GB1a	GB1a administration reverses loss of body weight and disease activity index scores in experimental colitis	[154]
	DSS-induced colitis	Nrf2 activation by dimethyl fumarate (DMF)	DMF attenuates the shortening of colons and alleviated colonic inflammation	[51,154,155]
	DSS-induced colitis	Nrf2 activation by Maresin 1 and Nrf2 inhibition by ML385	Maresin 1 attenuates experimental colitis by reducing activation of TLR4/NF-κB. ML385 reverses the protective effects of maresin 1 markedly	[157]
	4,6-trinitro-benzenesulfonic acid (TNBS) induced colitis	Nrf2 activation by Imperatorin	Imperatorin administration alleviates the symptoms of ulcerative colitis and inhibited the secretion of TNF-α and IL-6	[160]
	Acetic acid (AA)-induced colitis in rats	Nrf2 activation by Olmerartan	Olmerartan ameliorates colon injury and inflammatory signs	[161]
Systemic Lupus Erythematosus	Female Nrf2^−/−^ mice	Nrf2-deficient mice	Multiorgan inflammatory lesions Apearance of anti-double-stranded DNA antibodies in young adulthoodintravascular Pemature death due to rapidly progressing membranoproliferative glomerular nephritis	[167]
	B6/lpr mouse (sponatenous lupus nephritis model)	Nrf2-deficient mice	Nrf2 deficiency increases lupus nephritis and Th17 cell numners in B6/lpr mice	[66]
	MRL/lpr mouse	Nrf2-deficient mice	Nrf2 deficiency increases life span, improves nephritis. Immunologic abnormalities as well as hypergammaglobulinemia is correctetd.	[168]
	NZB/W mouse (spontaneous lupus nephritis model)	Nrf2 activation by A-1396076	A-1396076 dampens inflammation in an IFN-α-accelerated NZB/W mouse lupus nephritis model	[171]
	B6.Sle1.Sle3 mouse and MRL/lpr mouse	Nrf2 activation by CDDO-Me	CDDO-Me reduces severity of lupus disease by attenuating MEK-1/2, ERK, and STAT-3 signaling in CD4^+^ T cells, as well as oxidative stress in B6.Sle1.Sle3 mice or MRL/lpr mice	[172]
	Pristane-induced lupus nephritis mice	Nrf2 activation by DMF	DMF ameliorates pristane-induced lupus nephritis mice, and showes stronger anti-inflammatory and organ-protective effects than glucocorticoids	[173]
	Pristane-induced lupus nephritis mice	Nrf2 activation by sulphoraphane	Sulphoraphane suppresses pritane-induced lupus nephritis	[174]
Rheumatoid Arthritis	Complete Freund’s adjuvant-induced arthritis in rats	Nrf2 activation by DMF	DMF ameliorates complete Freund’s adjuvant-induced arthritis by suppressing oxidative stress and inflammatory mediators, and by increasing local Nrf2 and HO-1 concentration in the involved joints	[178]
	Collagen-induced arthritis in DBA/1 mice	Nrf2 activation by kurarinone	Kurarinone reduces arthritis severity of CIA mice, as well as their levels of proinflammatory cytokines in the serum and paw tissues	[182]
	Collagen-induced arthritis in DBA/1 mice	Nrf2 activation by oleuropein	Oleuropein containing diet prevents histological damage and arthritic score development	[183]
	Collagen-induced arthritis in DBA/1 mice	NR1D1 activation by SR9009 increases Nrf2-associated enzymes.	SR9009 significantly suppresses synovial hyperplasia, infiltration of inflammatory cells, and destruction of cartilage and bone in mice with CIA	[184].
	Collagen-induced arthritis in DBA/1 mice	Nrf2 activation by DC32, a dihydroartemisinin derivative	DC32 significantly alleviates footpad inflammation, reduce cartilage degradation	[185]

**Table 2 ijms-23-02846-t002:** Clinical trials using Nrf2 activators in metabolic/inflammatory diseases.

Compound Name [Mechanism]	Disease Target	ClinicalTrials.gov Identifier	Status	Phase
Synthetic compounds				
Bardoxolone methyl (CDDO-Me, BARD, RTA-402) [Electrophilic compunds]	Obesity	NCT04018339	Completed	I
	Hereditary nephritis (Alport syndrome)	NCT03019185	Completed	II/III
	Autosomal dominant polycystic kidney disease (ADPKD)	NCT03918447	Recruiting	III
	Pulmonary hypertension	NCT03068130	Terminated	III
	Connective tissue disease-associated pulmonary arterial hypertension	NCT02657356	Terminated	III
	Severe acute respiratory syndrome coronavirus 2 (SARS-CoV-2) infection; Coronavirus disease 2019 (COVID-19)	NCT04494646	Completed	II/III
	Focal segmental glomerulosclerosis (FSGS)	NCT03366337	Completed	II
	Diabetic kidney disease	NCT00811889	Completed	II
		NCT00550849	Terminated	I/II
		NCT00664027	Completed	II
		NCT03550443	Active, not recruiting	III
	Type 2 diabetes	NCT02316821	Completed	II
		NCT01053936	Completed	II
		NCT01053936	Completed	II
	CKD associated with type 1 diabetes	NCT03366337	Completed	II
	CKD associated with type 2 diabetes	NCT01351675	Terminated	III
	Chronic kidney disease	NCT04702997	Active, not recruiting	II
Dimethyl fumarate (Brand name Tecifidera^®^) [Electrophilic compunds]	Pediatric multiple sclerosis, relapsing-remitting	NCT03870763	Recruiting	III
	Multiple sclerosis	NCT02097849	Completed	II
	Ischemic stroke	NCT04891497	Not yet recruiting	II
	Obstructive sleep apnea	NCT02438137	Completed	II
	Pulmonary hypertension Scleroderma	NCT02981082	Terminated	I
	Age-related macular degeneration (AMD)	NCT04292080	Not yet recruiting	II
	Psoriatic arthritis	NCT02475304	Withdrawn	II
Oltipraz (CB-1400) [Electrophilic compunds]	Non-alcoholic fatty liver disease (NAFLD)	NCT04142749	Recruiting	II/III
Omaveloxolone [Electrophilic compunds]	Friedreich’s ataxia	NCT02255435	Active, not recruiting	II/III
	Mitochondrial myopathy	NCT02255422	Completed	II
	Cataracts	NCT02128113	Completed	II
	Ocular inflammation	NCT02065375	Completed	II
	Liver cirrhosis, liver failure	NCT03902002	Completed	I
Ursodiol (Ursodeoxycholic acid, brand names Actigall^®^ or Urso^®^) [Electrophilic compunds]	Parkinson’s disease	NCT03840005	Completed	II
	Ulcerative colitis	NCT03724175	Recruiting	II/III
	Type 2 diabetes	NCT02033876	Completed	II
	Hepatic steatosis	NCT03664596	Completed	II
	Retinopathy	NCT02841306	Completed	I
Dimethyl curcumin (AJ-101, ASC-J9) [Electrophilic compunds]	Acne vulgaris	NCT00525499	Completed	II
	Inflammatory acne	NCT01289574	Completed	II
AJ-201 (ALZ-002, ASC-JM-17) [Electrophilic compunds]	Spinal and bulbar muscular atrophy	NCT04392830	Completed	I
Natural compounds				
Sulforaphane (SFN) [Electrophilic compunds]	Type 2 diabetes	NCT02801448	Completed	II
	Cognitive disorders	NCT04252261	Not yet recruiting	II
	Chronic obstructive pulmonary disease (COPD)	NCT01318603	Completed	II
	Asthma	NCT00994604	Completed	NA
	Schizoaffective disorder, Schizophrenia	NCT02810964	Completed	II
	Autism spectrum disorders	NCT02654743	Completed	II
Sulforadex (SFX-01) [Electrophilic compunds]	Subarachnoid hemorrhage	NCT02614742	Completed	II
Curcumin [Electrophilic compunds]	Prediabetes	NCT03917784	Unknown	IV
	Diabetic nephropathy	NCT03262363	Unknown	II/III
	Type 2 diabetes	NCT02529969	Unknown	II/III
		NCT01052597	Unknown	IV
		NCT01052025	Unknown	IV
Resveratrol [Electrophilic compunds]	Diabetic nephropathy	NCT02704494	Completed	I
	Gestational diabetes	NCT01997762	Unknown	IV
	Type 2 diabetes	NCT01677611	Completed	I
		NCT01158417	Unknown	II/III
		NCT02244879	Completed	III
		NCT02216552	Completed	II/III
		NCT01354977	Completed	II
Quercetin [Electrophilic compunds]	Type 2 diabetes	NCT00065676	Completed	II
		NCT01839344	Completed	II

## Data Availability

Not applicable.

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
