# Peer review of "Recent Advances in Understanding Nrf2 Agonism and Its Potential Clinical Application to Metabolic and Inflammatory Diseases"

_ijms, 2022, doi:10.3390/ijms23052846_

Round 1
Reviewer 1 Report
Please see attached PDF.

Reviewer 2 Report
The article ‘Recent advances in understanding Nrf2 agonism and its potential clinical application in metabolic and inflammatory diseases’ by Kim and Jeon describes the role of transcription factor Nrf2 as a potential therapeutic target in several metabolic/inflammatory disorders with a focus on diabetic complications, obesity, Inflammatory bowel disease (IBD) and Systemic lupus erythematosus (SLE). The authors also discuss the ongoing clinical trials with Nrf2-related drugs. The review article is structured nicely, and the work is relevant for the readership of ‘IJMS.’ However, at some places, sentences are not crafted carefully and are worded rather casually. It becomes difficult for the reader to interpret the outcome. Some figure legends are also superficial. In addition, the article does not discuss open questions and the strategies to tackle them. The authors need to address these critical concerns in the current version of the manuscript before its publication. The major and minor issues are listed below.
Major:
In Figure 1 legend, Nrf2 autoregulation, post-translational modification, and stress-induced modification of cysteine thiols of Keap1 should be described. Further, there is no description of Keap1 structure either in text or in the figure legend.
It will be better to keep the blocks describing oxidative stress-related insufficient compensation of ROS pathophysiology out of the nuclear area in Figure2 and instead describe them outside the cell.
While the authors have discussed a clinical trial about CKD-associated type 1 diabetes, not enough literature on type1 diabetic complications and the role of Nrf2 is discussed.
Recent work related to the neuroprotective role of Nrf2 on diabetic encephalopathy and regulation of cell senescence should also be discussed.
The authors list several pro and anti-inflammatory cytokines regulated by Nrf2; however, a more mechanistic role of Nrf2 in regulating inflammatory enzymes such as MMPs, COX-2, iNOS, and pathways such as NF-ĸB are missing. Also, an up-to-date list of inflammatory diseases involving NRF2 misregulation and their pathophysiology should be provided in a tabular format.
The oxidative stress independent and Non-canonical Keap1-independent activation of Nrf2 is not discussed. While the authors may wish to focus on oxidative stress-related activation of Nrf2, these areas should be discussed briefly.
Please provide a summary of the inflammation related phenotypes discussed in the text in a table with references.
While the authors have attempted to summarize the relevant research in the field, they have not discussed any new hypothesis, open questions, or strategies to tackle them.
At some places, the manuscript lacks tense verb consistency. Further, the authors have used long, unclear sentences, often diluting the interpretation and conclusion. These should be taken care of in the revised version of the manuscript.
Minor:
Nrf2 contains seven domains named Nrf2-euthyroid cell-derived protein with CNC homology (Neh) domains (Figure 1).
There seems to be a typing error – euthyroid?
It will be helpful for the readers to label the amino acid numbers in the domain structures of Nrf2 and Keap1 in Figure1.
Keap1 acts as a sensor for Nrf2; under basal conditions, Keap1 binds to Nrf2, is ubiquitinated by the Keap1 Cul3 complex, and is then degraded by the 26S proteasome [40].
Rephrase this sentence. It appears Keap1 is ubiquitinated instead of Nrf2.
Thus, Nrf2 influences several important metabolic processes apart from antioxidant metabolism, including glucose and lipid metabolism, iron metabolism, and the anti-inflammatory response.
Include references here.
Did the authors mean NADPH production in Figure 2?
while in Treg cells, Nrf2-mediated antioxidant defense induces expansion and survival. By contrast, Nrf2 deficiency increases oxidative damage, which induces differentiation of Th17 cells, which are associated with a pro-inflammatory response. Finally, Nrf2 activation in myeloid-derived suppressor cells leads to expansion of MDSCs that inhibit T-cell responses
Please provide reference.
Although 5-aminosalicylic acids including mesalazine, steroids, and antagonists of TNF-α or other inflammatory cytokines are approved for treatment for IBD, its pathogenesis is still incompletely understood.
Please provide reference.
In line with this, activation of the ERK/Nrf2/HO-1 signaling cascade enhanced miophagy and thereby upregulated occludin and zona occluden proteins in the intestinal epithelial layer in a traumatic brain injury model [135].
There seems to be typing errors – Miophagy, zona occluden ?
Reviewer 3 Report
This review highlights the importance of NRF2 as a therapeutical target and covers the latest applications in the clinics. The review is good structured, and the information presented is of enough significance to merit its publication. Also, figures added are very useful to understand the information present. However, I have a few comments:
In section 2.2 The KEAP1-NRF2 system in regulating NRF2 authors claimed that regulation of NRF2 occurs through several mechanisms including ubiquitin proteasome degradation system (UPS), post-translational modifications epigenetic regulation, microRNAs, and autoregulation. However, in the text they explain only proteasome degradation. Considering that the other mechanisms for NRF2 regulations are less well-known, it would be interesting to detail these other regulatory pathways in this review.
Briefly describe the mechanism of action of the different inhibitors listed in Table 1.
NRF2 has emerged lately as an important target for the treatment of rheumatic diseases. Including a couple of sentences in the actual review will strength the content.
In section 5.1. NRF2 pharmacological activators under clinical trial, some of the drugs commented has been approved for treatment and are used in clinics. I suggest removing the title of this subsection and use the title of section 5. Also, in Figure 4 differentiate the drugs that are under clinical trials to the ones that have been already approved
Round 2
Reviewer 1 Report
The authors are to be commended for their substantial efforts to satisfy the reviewer comments, and on this comprehensive, well-referenced and in-depth review on Nrf2 in metabolic and inflammatory disease.
Reviewer 2 Report
The revised version of the manuscript 'Recent advances in understanding Nrf2 agonism and its potential clinical application in metabolic and inflammatory diseases' by Kim and Jeon has addressed all the relevant major concerns raised in the previous version of the paper. The manuscript is now fit for publication in 'International Journal of Molecular Sciences.' Hence, I endorse the publication of this paper in the journal.

Reviewer 3 Report
Authors addressed all my concerns.